# Automated Cardiac Chamber Size and Cardiac Physiology Measurement in Water Fleas by U-Net and Mask RCNN Convolutional Networks

**DOI:** 10.3390/ani12131670

**Published:** 2022-06-29

**Authors:** Ferry Saputra, Ali Farhan, Michael Edbert Suryanto, Kevin Adi Kurnia, Kelvin H.-C. Chen, Ross D. Vasquez, Marri Jmelou M. Roldan, Jong-Chin Huang, Yih-Kai Lin, Chung-Der Hsiao

**Affiliations:** 1Department of Chemistry, Chung Yuan Christian University, Taoyuan 320314, Taiwan; ferrysaputratj@gmail.com (F.S.); smalifarhan@gmail.com (A.F.); michael.edbert93@gmail.com (M.E.S.); kevinadik-adi@hotmail.com (K.A.K.); 2Department of Bioscience Technology, Chung Yuan Christian University, Taoyuan 320314, Taiwan; 3Department of Applied Chemistry, National Pingtung University, Pingtung 90003, Taiwan; kelvin@mail.nptu.edu.tw; 4Department of Pharmacy, Research Center for Natural and Applied Sciences, University of Santo Tomas, Manila 1008, Philippines; rdvasquez@ust.edu.ph; 5Faculty of Pharmacy, The Graduate School, University of Santo Tomas, Manila 1008, Philippines; mmroldan@ust.edu.ph; 6Department of Computer Science, National Pingtung University, Pingtung 90003, Taiwan; 7Center for Nanotechnology, Chung Yuan Christian University, Taoyuan 320314, Taiwan; 8Research Center for Aquatic Toxicology and Pharmacology, Chung Yuan Christian University, Taoyuan 320314, Taiwan

**Keywords:** water flea, deep learning, cardiac physiology, U-Net, Mask RCNN

## Abstract

**Simple Summary:**

With the rapid development of technology, artificial intelligent become a major breakthrough that can help human with laborious job. Previously cardiac imaging in *Daphnia* was also suffer from laborious and tedious process to extract some information from it. Thus the aim of this study was to develop a simple artificial intelligent based method to help anyone in this field to perform analysis in fast, reliable, and less tedious manner. In this study, we compare U-Net and Mask RCNN and found out that Mask RCNN was perform better than U-Net in cardiac chamber area estimation. From this data, several parameter like heart rhythm, stroke volume, ejection fraction, fractional shortening, and cardiac output can be extracted. The validation was done by comparing the normal and Roundup exposed group and it show that Roundup can increase the stroke volume, cardiac output, and the shortening fraction of *Daphnia magna.*

**Abstract:**

Water fleas are an important lower invertebrate model that are usually used for ecotoxicity studies. Contrary to mammals, the heart of a water flea has a single chamber, which is relatively big in size and with fast-beating properties. Previous cardiac chamber volume measurement methods are primarily based on ImageJ manual counting at systolic and diastolic phases which suffer from low efficiency, high variation, and tedious operation. This study provides an automated and robust pipeline for cardiac chamber size estimation by a deep learning approach. Image segmentation analysis was performed using U-Net and Mask RCNN convolutional networks on several different species of water fleas such as *Moina* sp., *Daphnia magna*, and *Daphnia pulex*. The results show that Mask RCNN performs better than U-Net at the segmentation of water fleas’ heart chamber in every parameter tested. The predictive model generated by Mask RCNN was further analyzed with the Cv2.fitEllipse function in OpenCV to perform a cardiac physiology assessment of *Daphnia magna* after challenging with the herbicide of Roundup. Significant increase in normalized stroke volume, cardiac output, and the shortening fraction was observed after Roundup exposure which suggests the possibility of heart chamber alteration after roundup exposure. Overall, the predictive Mask RCNN model established in this study provides a convenient and robust approach for cardiac chamber size and cardiac physiology measurement in water fleas for the first time. This innovative tool can offer many benefits to other research using water fleas for ecotoxicity studies.

## 1. Introduction

Cladocera, commonly known as water fleas, belong to the subclass Phyllopoda of the class Crustacea [1]. They are lower invertebrate animals often shaped like flat disks with small sizes ranging from 0.2 to 3 mm in length. They are widely distributed in the global freshwater ecosystem and have a vital role in the aquatic food chain as one of fish’s primary natural food sources [2]. Cladocera is an essential component of microcrustacean zooplankton, which are sensitive indicators of environmental changes. They provide early warnings by demonstrating a prompt response to environmental changes [3].

Most of the order Cladocera belongs to the suborder Anomopoda, which is principally comprised of the families Daphniidae (the Genus *Daphnia*) and Moinidae (Genus *Moina*) [1], which are the common water flea crustacea used for ecotoxicological testing [4,5,6,7]. With sufficient food and aeration, they are easily cultured in the laboratory. High availability, rapid reproduction, and economic feasibility are the reasons for selecting this animal model for a toxicity test [8]. They have high sensitivity toward various chemical pollutants found in the environment, such as metals, pesticides, and pharmaceuticals [9,10,11,12]. The Organization for Economic Cooperation and Development (OECD) has described the international guidelines of standard bioassays using freshwater cladocerans to determine the toxicity of chemicals and pollutants [13]. The ecotoxicological assessment is based on easily measurable endpoints such as lethality, growth, reproduction, immobilization, or behavior [14]. Another important parameter, cardiovascular function, is also usually used as an indicator of toxicity evaluation. A previous study displayed that the water flea’s heartbeat, cardiac output, and heartbeat regularity are significantly reduced when exposed to the pesticide imidacloprid [15]. It is plausible to detect and measure cardiovascular performance with their body transparency, which is suitable for ecotoxicity assay [16,17].

Digital imaging processing (DIP) is a well-known subject for analyzing image datasets [18]. Unlike task-based methods, deep learning (DL) comprises machine learning data processing [19]. It has efficient results, especially on larger datasets. The big data world is using DL for many of its applications. It has been considered that neural networks are significantly designed to converge computer vision programs with DL applications. The traditional approaches in DL are used for video sequencing and image processing. Digital data acquisition is a common practice in medical informatics with the help of DL. Heart chamber size measurement and volumetric behaviors are established using the DL models. Two-dimensional convolutional neural networks (CNN) are the principle neural network extension that converges pixel-based instant segmentation. Kernel slides with 2D networks convolve the layer to the previous with height and width, which is used to extract the features of the image. The Deep Convolutional Neural Networks (DCNNS) are integrated with high-level performance in image classification with soaring heights array [20].

Good classification and segmentation algorithms always help researchers find accurate test cases in the desired image. In the field of computer vision, several different methods such as U-Net [21,22], Mask RCNN [23], and YOLO [24,25] have been developed in the past decade, which put the image classification world in more conducive manners [26]. In previous studies conducted by Karatzas et al. [27], the Mask RCNN method has been tested to detect heart malformation in *Daphnia*. Zhao et al. also utilized the U-Net architecture to segment cardiac chamber in magnetic resonance images [28]. A further study by Dong et al. showed that U-Net could also be used for video segmentation for cardiac MRI video with state-of-the-art performance [29]. All those studies have proven that the deep learning approach was beneficial for cardiac chamber segmentation.

Previous studies measured the cardiac performance endpoints in water fleas based on image-based methods (Table 1). Multiple endpoints such as heart rate, cardiac output, ejection fraction, fractional shortening, and heartbeat regularity can be extracted. Heart size, contraction capacity, and heart shape malformation due to toxicants have also been studied [17]. All these endpoints were achieved based on the manual analysis of heart images and calculating the cardiac chamber size during heart relaxation (diastole) and contraction (systole) [30]. Therefore, cardiac physiology, as well as morphology observation, is fundamental in cardiac performance evaluation. Both quantitative and qualitative methods are essential to detect possible alterations of heart function in this animal model due to exposure to hazardous or toxic substances within the ecotoxicology field. Based on Table 1, we learned that most of the previous cardiac physiology measurement methods were conducted using manual counting or ImageJ-based methods. However, slow and complex operation processing is the major drawback that makes previous methods unsuitable for high-throughput toxicity assessment. Thus, the incorporation of deep learning will really benefit everyone in this feld of study as a fast and reliable performance can be achieved with help of machine learning.

As mentioned above, the drawback of the previous assessment method was the slow and complex operation. In addition, it is also prone to human error as most of the process was conducted by human operator. Thus, the main focus of this study was to develop a machine learning based tool to undertake the assessment of cardiac physiology in water fleas. This study will compare the performance of U-Net and Mask RCNN on cardiac image segmentation for water fleas after being trained with a video of water fleas’ heart chamber. Later the optimized convolutional networks will be used to predict the cardiac chamber size of water fleas, and from this, data several cardiac physiology parameters can be obtained. The successful establishment of this novel approach can significantly boost the throughput for cardiac physiology and toxicity studies in water fleas while also reducing the possibility of error caused by human mistake.

## 2. Materials and Methods

### 2.1. Water Flea Culture

This study used three water flea species *Daphnia magna*, *Daphnia pulex,* and *Moina* sp. Cultures of *D. magna* and *D. pulex* were obtained from the National Chiayi University stock center, and *Moina* sp. was obtained from the National Pingtung University stock center. The water fleas were kept in 10 L plastic tanks and supplied with baker yeast as food, and the temperature was maintained at ±24 °C. To maintain the healthy culture conditions, half of the old culture water was removed weekly and replaced with freshwater.

### 2.2. Chemical Exposure

The roundup (41% *w*/*v*) was purchased from Yih Fong Chemical Corp. (Taichung, Taiwan) and then diluted into 1000 ppm stock concentration using ddH_2_O and kept at 4 °C until the time of exposure. At the time of exposure, *D. magna* was placed into a 5 cm petri dish. The stock solution was further diluted using *Daphnia* culture water until the concentration of 5 ppm. The exposure was performed for 24 h. The *Daphnia* culture water was used to reduce the shock due to sudden water temperature change caused by the different water used for the experiment. All protocols and procedures involving *Daphnia* were approved by the Committee for Animal Experimentation of the Chung Yuan Christian University (Approval No. 109001, issue date 15 January 2020).

### 2.3. Video Acquisition

For cardiac chamber size estimation, the videos of water fleas were captured using a high-speed charged coupled device (CCD) camera (AZ Instrument, Yuyao, Taiwan), mounted onto an inverted microscope (Sunny Optical Technology, Yuyao, China). The LPlan objective lens with 20× magnification was used to capture the video at high quality. Videos were recorded for 10 s at a frame rate of 200 frames per second (fps) and later converted to 30 fps to create slow-motion videos with a total of 2000 frames using HighBest Viewer software (AZ Instrument, Taichung City, Taiwan). For the mounting solution, 3% methylcellulose was used to immobilize water fleas before recording the cardiac chamber.

### 2.4. Training Dataset Preparation

After recording, the video format was converted into .avi format with VirtualDub software (Available online: http://www.virtualdub.org/, accessed on 26 June 2022) and then processed using ImageJ [39]. A total of ten frames were selected from the video as the dataset and outputted in .png format at an interval of 200 frames for each video. The training dataset was prepared by marking the image extracted from the video using ImageJ. The border of the heart chamber was marked with pencil tools using ImageJ, and the marked images were saved in .png format (Figure A1). When the image set was not big enough, image augmentation was used to increase the size of the training image [40,41]. The image augmentation methods performed included cropping, flipping, grid distortion, elastic transform, optical distortion, and brightness contrasting [42]. In total, 600 images were included as a training dataset, and manual labeling was performed on the given dataset. The training dataset was used separately to train both Mask-RCNN and U-Net models. The code used for 

### 2.5. Performance Validation

The four following parameters were used for deep learning performance evaluation: Dice coefficient, Intersection over Union (*IOU*), sensitivity, and specificity. Given the number of true positives (*TP*), false positives (*FP*), and false negatives (*FN*) in the pixel-wise classification of the predicted mask, the Dice coefficient is defined as $\mathrm{Dice}=\frac{2TP}{2TP+FP+FN}$. The *IOU* is defined as $IOU=\frac{TP}{TP+FP+FN}$. The sensitivity (also called *recall*) is defined as $\mathrm{sensitivity}=\frac{TP}{TP+FN}$. The specificity is defined as specificity = $\frac{TN}{TN+FP}$. Those four important parameters were collected and measured for the U-Net and Mask RCNN methods.

### 2.6. Volumetric Estimation of Water Flea’s Heart

In calculating the volume of the water flea heart chamber, the Open source Computer Vision (OpenCV) library was used for ellipse fitting and point detection of the long and short axis of the water flea’s heart chamber [43]. The black-white frame-by-frame image exported from Mask-RCNN was used as the basis for volumetric assessment to reduce the background noise present in the original image. The basic concept was initiated by finding contour boundaries using Python’s fit ellipse function [44]. Contour is defined as the line surrounding the water flea’s heart chamber. Contour can be presented in multiple ways as polygons and Freeman chain codes [45]. In OpenCV, contours are mostly computed using binary images as it was easier to define contrast in respective images. Once contour was computed, the next step was to fit an ellipse with a diameter range of the water flea’s heart chamber. Cv2.fitEllipse function in OpenCV was used to estimate the x and y coordinates of the ellipse diameter [46]. Finally, the short and long axes that fit the heart chamber area were extracted and exported into .csv format automatically in the respective folder (Figure A2). 

### 2.7. Computer Hardware Requirement

The proposed experimental design was implemented using the deep learning library Pytorch [47] on a desktop computer running the Linux operating system with AMD Ryzen 9 5900X Computer Processing Unit (CPU), 128- gigabyte Random Access Memory (RAM), 64 Terrabyte hard disk storage, and RTX3090 24G VRAM Graphic Processing Unit (GPU). Although a lower computer specification could also be used for the study, a high-speed GPU card is necessary to complete the design and testing of the model faster.

### 2.8. Cardiac Performance Analysis

In validating this newly developed method in detecting the cardiac rhythm, the heart rate and the heart rate variability were calculated and then compared with the previously published ImageJ method [15]. In calculating the heart rate using the newly developed method, the size of the heart chamber from time to time was exported and then processed using OriginPro 2019 software (Originlab Corporation, Northampton, MA, USA). The timing of the heart muscle relaxation was extracted using the OriginPro 2019 software, and the average time interval between each relaxation was calculated. Later the heart rate per minute was calculated by dividing 60 with the average time interval. Calculating the heart rate using the method was performed by detecting the difference in brightness at the Region of Interest (ROI). In this case, the heart chamber using Time Series Analyzer V3 Plugin is available in ImageJ software (https://imagej.nih.gov/ij/plugins/time-series.html, accessed on 26 June 2022). Later the data were also processed similarly by using OriginPro 2019 software to get the timing of heart muscle relaxation. The Poincare Plot Plugin on OriginPro 2019 software was used to calculate the heart rate variability, and the sd1 and sd2 generated were noted and compared statistically. In calculating the cross-section area change, the following formula was used: $\frac{EDV\;area-ESV\;area}{EDV\;area}\times 100\%$. (*EDV*, end diastolic volume; *ESV*, end systolic volume). Other cardiac physiology parameters such as stroke volume, cardiac output, shortening fraction, and ejection fraction were calculated using the same concept and formulas described in the previous study [15].

### 2.9. Statistical Calculation

Statistical analysis was performed using GraphPad Prism (GraphPad Inc., La Jolla, CA, USA). Depending on the data distribution, a paired *t*-test or Wilcoxon test was performed to calculate the statistical significance between the method used, and a student *t*-test and a Mann–Whitney test was performed to calculate the significance between the control and the treatment group [48,49].

## 3. Results

### 3.1. Overview of Experimental Design and Training Dataset Preparation

In this study, the collection of water flea heartbeat videos was captured by using high-speed CCD. The high-speed CCD setup with 200 fps frame rate can capture superfast heartbeat in water fleas with less information loss [15]. Later, images were frame-by-frame outputted from the video, and the heart’s outlooking was manually labeled as a training dataset. Two predictive models of mrcnn_predict.py (source code is provided in Appendix A) and unet_predict.py (source code is provided in Appendix A) were established after extensively training by Mask RCNN or U-net methods, respectively. The performance of both predictive models was tested to predict the heart size (area) of the water flea. Once we got the heart area prediction data, the major and minor axis length could be extracted using the Cv2.fitEllipse function (axis.py, source code is provided in Appendix A) in OpenCV. Finally, cardiac physiology endpoints such as cardiac rhythm, heartbeat regularity, heart 2D/3D area, cardiac output, stroke volume, fractional shortening, and ejection fraction, could be obtained by using the mathematic calculation function in excel (the entire experimental design is summarized in Figure 1). 

### 3.2. Training and Validation Performance

We evaluated deep learning’s performance by comparing the dice and loss curves in the training and validation processes. In order to train our U-Net and Mask RCNN models, we prepared 540 *D. magna* images and their ground truth segmentation images as the dataset. In the training stage, 60 of the 540 images in the dataset were used as the validation set, and the remaining 480 images were used as the training set. Each image in the validation and training set was augmented into 15 images using image processing methods such as affine transform, rotation, flip, and deformation. Therefore, there are 7200 images for training and 900 images for verification. For training by the U-Net method, the dice coefficient increased exponentially from the first steps and reached the maximal plateau after about 4300 steps. The loss curve reached the minimal level after about 3000 steps. The dice coefficient reached the maximal plateau after about 3600 steps for validation. Similarly, the loss curve showed a minimal level of about 3600 steps (Figure 2A). For validation by the Mask RCNN method, the dice coefficient gradually increased starting from the first step and reaching the maximal plateau after about 25,000 steps. The loss curve also appeared from the first step and reached the minimal level after 40,000 steps (Figure 2B). The high dice coefficient for both training (Figure 2C) and validation (Figure 2D) processes suggests that the trained U-Net and Mask RCNN might predict heart size for *D. magna* with high accuracy. 

### 3.3. Testing Process

To test whether the optimized training networks for either U-Net or Mask RCNN are suitable for heart size prediction in water fleas, we initially used 60 images from *D. magna* as a testing dataset. We also determined whether the optimized networks have broad application utility to predict heart chamber size for other water flea species. Four parameters in terms of Dice coefficient, *IOU*, sensitivity, and specificity were used for performance evaluation, given the number of true positives (*TP*), false positives (*FP*), and false negatives (*FN*) in the pixel-wise classification of the predicted mask. Figure 3 shows a comparison of the prediction power between Mask RCNN (Figure 3A) and U-Net (Figure 3B). The prediction results were consistent with the ground truth in three tested water fleas using Mask RCNN. On the contrary, only *D. magna* showed consistent results between prediction and ground truth by the U-Net method. The other two closely related water fleas of *D. pulex* and *Moina* sp. displayed inconsistent ground truth and prediction results. Therefore, Mask RCNN’s performance is better than U-Net for heart size prediction in water fleas.

Next, we conducted a detailed quantitative comparison of the heart size prediction performance between Mask RCNN and U-Net among three water fleas in terms of Dice coefficient, *IOU*, sensitivity, and specificity. Results showed that the optimized Mask RCNN model trained by *D. magna* performed well and predicted heart chamber size with good performance for *D. pulex* and *Moina* sp. based on the high score obtained from Dice coefficient, *IOU*, sensitivity, and specificity (Table 2). On the contrary, U-net only displayed a qualified prediction power on *D. magna*, and relatively low prediction power on *D. pulex* and *Moina* sp. For example, based on *IOU*, Mask RCNN maintained relatively high scores of 0.940 ± 0.015, 0.919 ± 0.020, and 0.874 ± 0.083 for *D. magna*, *D. pulex,* and *Moina* sp., respectively. For U-Net, the *IOU* scores sharply declined from 0.872 ± 0.070 for *D. magna* to 0.707 ± 0.161 for *D. pulex* and 0.526 ± 0.228 for *Moina* sp. Therefore, the optimized Mask RCNN model established in this study can be applied to water flea’s heart size prediction with high accuracy which is supported by statistical analysis.

### 3.4. Analysis of Heart Cardiac Size Change over Time in Three Water Fleas by Mask RCNN

Since the trained Mask RCNN networks displayed good performance on heart size prediction in three tested water fleas of *D. magna*, *D. pulex,* and *Moina* sp., we evaluated whether it can be applied to measure cardiac rhythm and cardiac size change over time. We first outputted a 10 s high-speed video at 200 fps frame-by-frame into the image series to reach this goal. Later, 2000 frames were output as a testing dataset to perform heart size prediction using Mask RCNN. Finally, the cardiac rhythm plot was generated by plotting heart size as the y-axis and time as the x-axis. Using this approach, we found that the trained Mask RCNN network works as a universal tool for heart size prediction and cardiac rhythm detection for all three water flea species tested in this study (Figure 4). By analyzing the heart chamber size variation over time, it was able to recapitulate heartbeat rhythm for three tested water fleas. For example, *D. magna* (455 bpm, pink color) and *D. pulex* (460 bpm, green color) were found with a similar level of heartbeat rate which was faster than that detected in *Moina* sp. (208 bpm, blue color) (Figure 4). Videos for heart chamber prediction of *D. magna*, *D. pulex*, and *Moina* sp. can be found in Appendix A.

### 3.5. Validation of Cardiac Physiology Alterations in D. magna after Herbicide Exposure

We performed data validation for our newly developed deep learning method and previously reported ImageJ method for the next step. The result showed that no significant difference in heart rate was found in both the control (*p* = 0.9677) and roundup (*p* = 0.4982) treatment between deep learning and ImageJ-based methods (Figure 5A). However, after comparing the cross-sectional area change using our developed method and manual counting using ImageJ, we found that the deep learning method predicted heart size was significantly larger than the manual method conducted by ImageJ for both the control (*p* = 0.003) and Roundup treatment (*p* = 0.0125) (Figure 5B). We estimated they were around 6–8% bigger when the deep learning method was conducted. In the case of heart rate variability, no significant different was found in the sd1 (Control *p* = 0.2026 & Roundup *p* = 0.5144) (Figure 5C) and sd2 (Control *p* = 0.4458 & Roundup *p* = 0.3119) (Figure 5D) in both the treatment groups. These results strongly suggest that the newly developed deep learning method can give similar results to the already established ImageJ method except in the cross-sectional area change.

The last step was performed by analyzing other cardiac physiology parameters such as the stroke volume, ejection fraction, shortening fraction, and cardiac output by using the ellipse fitting from the OpenCV and point detection to get the distance of the long and short axes of the heart chamber. After getting the distance of both chambers, the calculation was performed by assuming the heart chamber has a three-dimensional ellipsoid shape, and the body size was normalized the volume to get a more accurate measurement of the heart pumping ability [15]. In this study, we observed a significant increase in the stroke volume (*p* < 0.0001), cardiac output (*p* < 0.0001), and the shortening fraction (*p* < 0.0078) of *D. magna* after incubation of 5ppm Roundup (Figure 6A,B,D). However, no significant difference was observed in the ejection fraction after incubation in 5 ppm of Roundup compared to the control, which suggests that at 5 ppm concentration, Roundup can increase the heart size of *D. magna* without significantly changing the pumping capability of the heart chamber.

## 4. Discussion

Several methods currently available in the literature may be suitable for marking the edge of the heart chamber. In this study, U-Net [21,22] and Mask RCNN [23] methods were adopted to conduct cardiac size prediction in water fleas as those two tools have been successfully reported to perform heart segmentation in humans [50,51,52,53]. We discovered that the heart chamber detection in water fleas using Mask RCNN shows superior performance in comparison to U-Net. U-net is a generic deep-learning solution for image detection and segmentation and can be used for biomedical image data analysis. U-Net uses the U-shaped network structure first to capture the features of the images and reconstruct the required partitions based on these features [22]. The U-Net architecture contains a U shape path. The image enters one end of the U shape network to go through the encoding part of the network (also called the contracting path). The encoding part is used to capture the features of the input images, and during this step, the spatial information is reduced while the feature information is increased. Then the information of the input image is sent to the decoding part (also called the expansive path). At the expansive path, the feature collected during the contracting step was combined through a high-resolution sequence of up-convolutions and concatenations to construct the segmentation image [22]. Therefore, U-Net’s method of fusing low-level and high-level image features may have a chance to mark the heart chamber’s edge successfully.

Unlike U-Net, Mask RCNN was used to solve instance segmentation problems in computer vision. Mask RCNN consists of two stages. The first step is to propose the region which contains the object based on the input image. Then in the second stage, it will predict the class of object, make the boundaries of the object, and generate a mask at pixel level based on the proposal from the first stage [54]. Compared to U-Net, which performs better on semantic segmentation, Mask RCNN performs better than U-net, for instance, in segmentation [55]. Several studies suggest that the combination of Mask RCNN and U-Net can outperform a single component only, which could be explored more to increase the accuracy of the image segmentation [56,57,58].

Our training result suggests that the Mask-RCNN model in this study performs better than the U-Net model in defining the heart chamber boundary. A similar case happened when both models were challenged to perform nuclei segmentation, which shows that Mask-RCNN has better precision than U-Net [59]. Another study also reported that U-Net yielded more false-positive results when detecting the immunofluorescence images of kidney biopsies from lupus nephritis patients than Mask-RCNN [60]. Although U-net has higher accuracy, the limitation of U-Net observed in the study was the obstruction of detection when labeled data has edge noise, thus yielding false-positive segmentation [59,61]. However, it can be said that both U-Net and Mask-RCNN models might have similar outputs if the training dataset has more viable point boundary determination. In conclusion, the accuracy and effectiveness of each model depend upon the targeted images that are being classified and segmented.

Another thing that could be observed in this study is the significant difference between manual counting using ImageJ and the Mask RCNN-based method to calculate the cross-section area change of the cardiac chamber. This might be because by using Mask RCNN, the calculation of the cross-sectional area change was based on the maximum and minimum cross-sectional area of the heart chamber from the whole cardiac cycle recorded in the video. On the contrary, the calculation was only based on randomly selected images from a few cardiac cycles for manual counting. Thus, the calculation using manual counting has a lower cross-sectional area change than the automated Mask RCNN method.

It is also worth noting that some limitations were noticed in this deep learning-based study. As the developed tool was based on detecting the edge of the heart chamber, the clarity of the heart chamber edge plays a crucial role in the accuracy of cardiac chamber prediction. In the case of *Daphnia* and *Moina* sp., the brood chamber was located right beside the heart chamber. Thus if the samples are in the breeding period, the accuracy of the tools would be compromised when the brood was positioned overlapping the heart chamber (Figure A3 & Appendix A) [62]. Another type of limitation is the presence of some parasites near the heart chamber. This problem could also reduce the clarity of the heart chamber, making it harder for deep learning to analyze the heart chamber [63]. Compared to the ImageJ-based method in which the selection of ROI can be made anywhere as long as it has distinct dynamic pixel change, our tool only works accurately if the whole heart chamber can be detected. However, this problem can be overcome by selecting the water fleas with an empty brood chamber or those which are parasite free, ensuring that heart chamber edge is undisturbed.

## 5. Conclusions

In conclusion, it can be said that Mask-RCNN performed better than U-Net in estimating *D. magna* cardiac chamber size. Higher Dice coefficient, specificity, and sensitivity show that the Mask-RCNN model was superior in *Daphnia* cardiac chamber segmentation. Furthermore, the segmentation result could also be used to calculate several cardiac performance parameters such as heart rate, heart rate variability, and cross-sectional area change, while with the addition of the ellipse fitting function from OpenCV, more parameters such as stroke volume, cardiac output, shortening fraction, and ejection fraction could also be calculated. Several limitations such as the dependency on the heart chamber clarity and the presence of obstruction nearby the heart chamber could decrease the accuracy of the segmentation, but both problems could be solved by selecting and sorting the *Daphnia* before recording the heart chamber. Overall, this study suggests that deep learning could help analyze cardiac physiological parameters in *Daphnia*. Although in this study, fully automated analysis was not yet achieved, in the future, fully automated high throughput analysis could be achieved by incorporating essential function in several software by using our trained network as the foundation.

## Figures and Tables

**Figure 1 animals-12-01670-f001:**
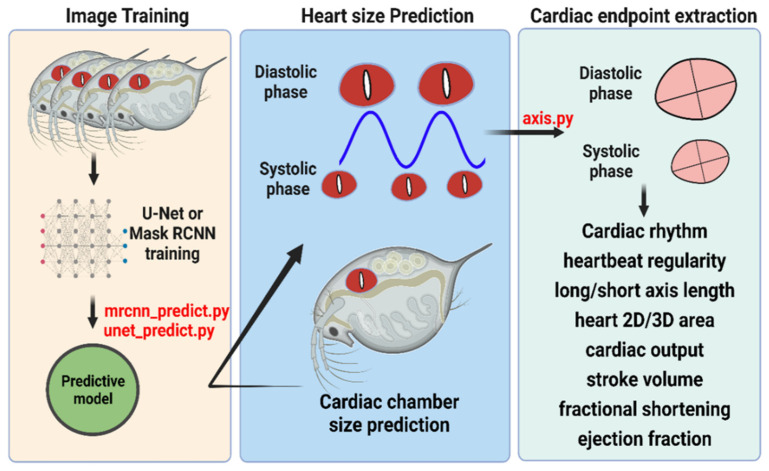
Experimental workflow for water flea heart size prediction by using a deep learning approach. Two different convolutional networks of U-Net and Mask RCNN were used for comparison.

**Figure 2 animals-12-01670-f002:**
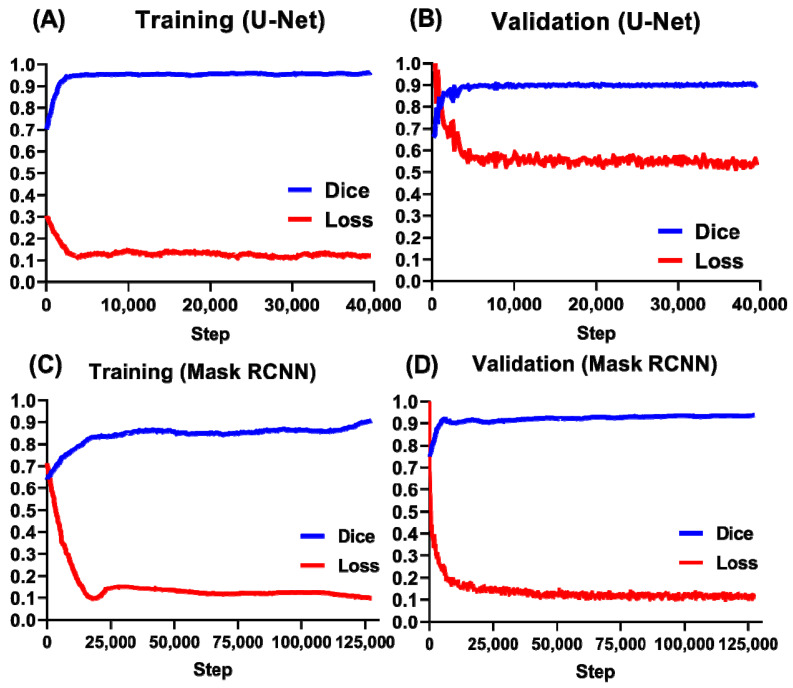
The Dice and loss curves for the U-net and Mask RCNN method for *D. magna* heart size prediction. The Dice and loss curve for the U-Net method at either training (**A**) or validation (**B**) process. The Dice and loss curve for the Mask RCNN method at either training (**C**) or validation (**D**) process.

**Figure 3 animals-12-01670-f003:**
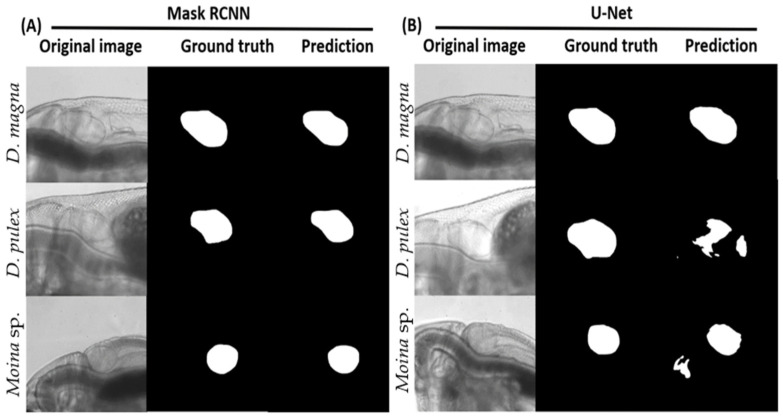
Image segmentation done using Mask RCNN (**A**) and U-Net (**B**) to predict heart size in water fleas. The white area show the predicted position of heart chamber. Three water flea species of *D. magna*, *D. pulex,* and *Moina* sp. were tested and the heart size for ground truth, and prediction is shown for comparison.

**Figure 4 animals-12-01670-f004:**
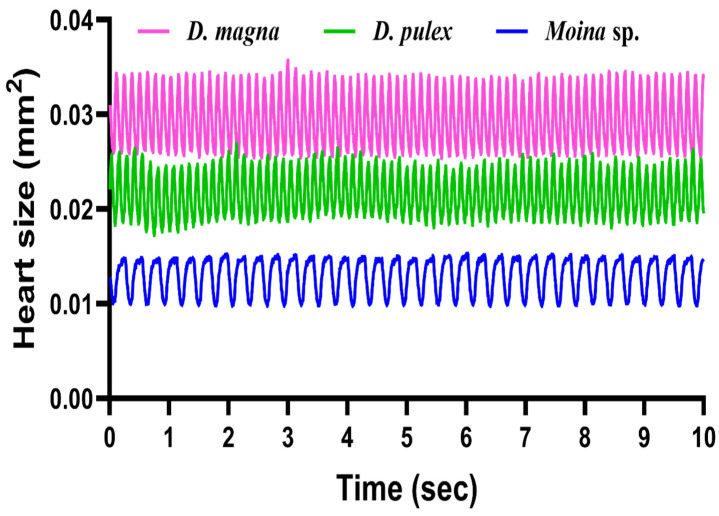
Use Mask RCNN to study cardiac rhythm in water fleas. Three water flea species of *D. magna* (pink color), *D. pulex* (green color), and *Moina* sp. (blue color) were tested, and the cardiac rhythm and heart size change dynamic can be elucidated by the Mask RCNN method.

**Figure 5 animals-12-01670-f005:**
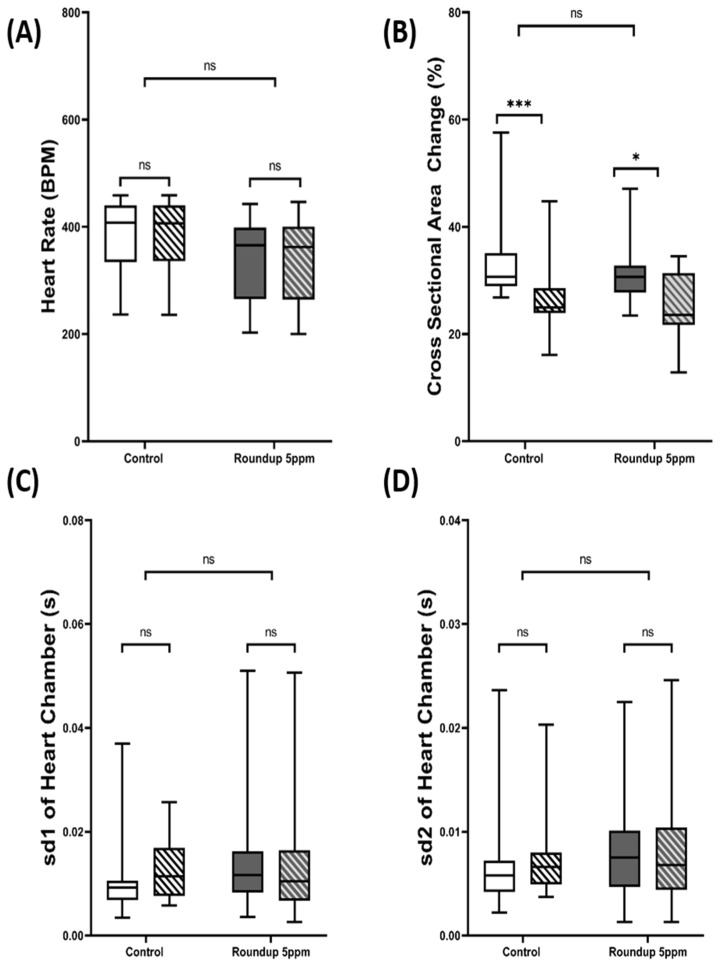
Comparison of cardiac physiology parameters after incubation in 5 ppm Roundup for 24 h using either the Mask RCNN or ImageJ method. The data were shown as a box and whisker plot with mean ± min to max values. The statistical significance was compared using either paired *t*-test (**A**) or Wilcoxon test (**B**–**D**) to analyze the intra-treatment result and using *t*-test (**A**) and Mann–Whitney test (**B**–**D**) to analyze the inter-treatment result (* *p*<0.05, *** *p*<0.001). The open box represented the Mask RCNN method, and the shaded box represented the ImageJ method.

**Figure 6 animals-12-01670-f006:**
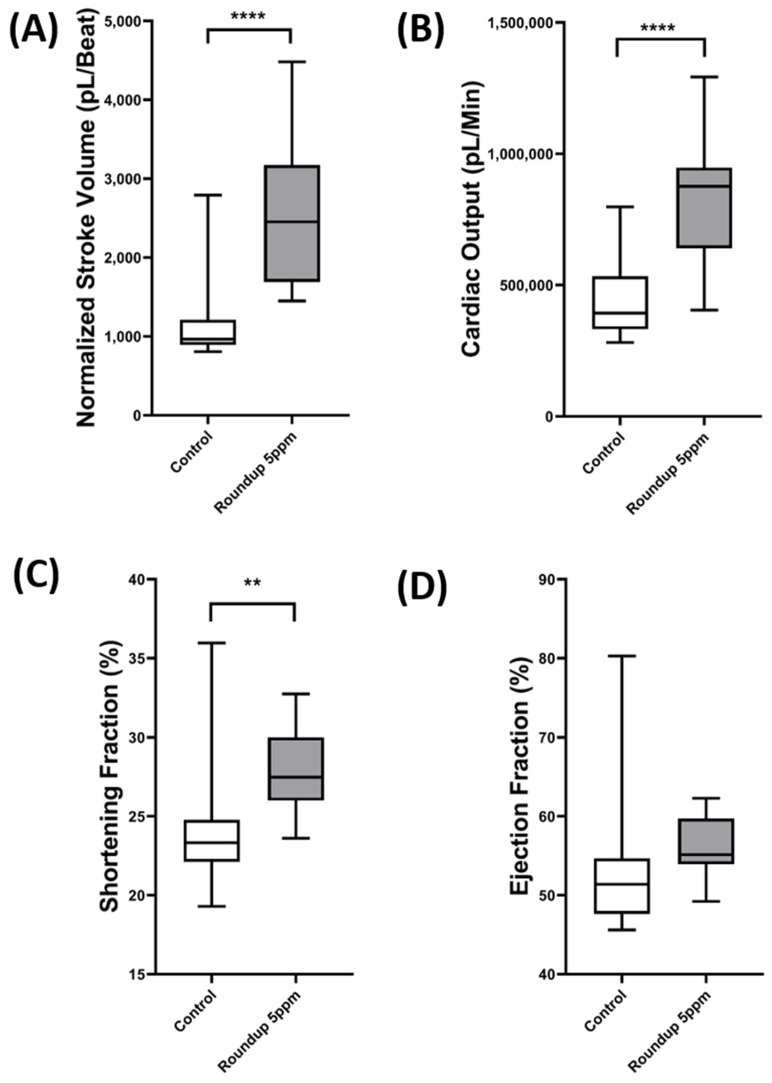
Comparison of cardiac physiology parameters after incubation in 5 ppm of Roundup for 24 h. The heart size was first predicted by Mask RCNN, and later the long and short axes of the cardiac chamber was predicted by OpenCV. Next, we used long and short axis lengths to assess cardiac physiology by measuring stroke volume (**A**), cardiac output (**B**), shortening fraction (**C**), and ejection fraction (**D**). The data were shown as mean ± SD, and the statistical significance was compared using an unpaired student *t*-test (** *p*<0.01, **** *p*<0.0001).

**Table 1 animals-12-01670-t001:** Comparison of previous cardiac physiology measurement methods in Cladocera.

Reference	Recording Instrument	Software/Tools	Animal model	Obtainable Result
This study	High-speed CCD camera mounted to an inverted microscope	U-Net and Mask RCNN convolutional Networks	*D. magna*, *D. pulex*, and *Moina* sp.	Cross sectional area change, heart rate, stroke volume, ejection fraction, fraction shortening, cardiac output, and heartbeat regularity
[31]	Spencer microscope devised with stroboscope	Stroboscope or stopwatch for manual counting with the naked eye	*D. magna*	Heart rate
[32]	Inverted microscope, digital video camera, and videotape recorder assembled to computer	Echocardiography	*D. magna*	Irregularity of cardiac rhythm, cardiac area in systole/diastole, and beats per min.
[33]	Digital camera attached to a microscope	Manual counting	*D. pulex*	Heart rate
[34]	Panasonic DMC-LZ8 camera	Movie maker was used to play the recording video in slow motion, then manual counting (beats/min) was conducted	*Simocephalus vetulus*	Heart rate
[35]	a digital camera Nikon D3100 mounted on a microscope.	Tracker® software	*D. magna*	Heart rate, diastole/systole heart area ratio, duration of diastole
[36]	microscope (CKX41SF, Olympus) equipped with a digital camera	GOM player and ImageJ software	*D. magna*	Heart size, contraction capacity, and heart rate
[27]	Nikon stereomicroscope, model SMZ800 Digital Sight, fitted with a D5-Fi2 camera	Image capture by NIS-Elements software and image analysis by machine learning (R-CNN)	*D. magna*	Heart malformation detection
[15]	High-speed CCD camera mounted to an inverted microscope	ImageJ Time Series Analyzer plug-in	*D. magna*, *D. similis*, and *Moina* sp.	Heart rate, blood flow rate, stroke volume, ejection fraction, fractional shortening, cardiac output, and heartbeat regularity
[37]	High-speed CCD camera mounted to an inverted microscope	ImageJ Kymograph plug-in	*D. magna*	Heart rate, stroke volume, ejection fraction, fraction shortening, cardiac output, and heartbeat regularity
[38]	High-speed CCD camera mounted to an inverted microscope	OpenCV	*D. magna*	Heart rate and heartbeat regularity

**Table 2 animals-12-01670-t002:** Comparison of prediction power of U-Net and Mask RCNN for cardiac size prediction in different water flea species.

	Dice Coefficient	*IOU*	Sensitivity	Specificity	N
**U-Net**					
*D. magna*	0.930 ± 0.042	0.872 ± 0.070	0.946 ± 0.084	0.987 ± 0.009	60
*D. pulex*	0.817 ± 0.124	0.707 ± 0.161	0.804 ± 0.173	0.989 ± 0.006	100
*Moina* sp.	0.659 ± 0.209	0.526 ± 0.228	0.732 ± 0.207	0.970 ± 0.029	100
**Mask RCNN**					
*D. magna*	0.969 ± 0.008	0.940 ± 0.015	0.967 ± 0.019	0.995 ± 0.005	60
*D. pulex*	0.958 ± 0.011	0.919 ± 0.020	0.945 ± 0.025	0.998 ± 0.001	100
*Moina* sp.	0.930 ± 0.054	0.874 ± 0.083	0.961 ± 0.032	0.994 ± 0.009	100

## Data Availability

The data presented in this study are available on request from the corresponding author.

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
