# Peer review of "Automated Cardiac Chamber Size and Cardiac Physiology Measurement in Water Fleas by U-Net and Mask RCNN Convolutional Networks"

_animals, 2022, doi:10.3390/ani12131670_

Round 1

Reviewer 1 Report

This manuscript explored the automated cardiac chamber size and cardiac physiology of water fleas which are important invertebrates.

The authors performed well in the experimental design.

The authors also mentioned the previous reports and the development of the cardiac physiology measurement of Claocera. This could help more understanding of the history and development of cardiac physiology measurement of Claocera.

For my point of view, it will be better if you can add more details of the impact of this work, such as more application using this automate. 

Moreover, the authors have to check the correction of Mask RCNN and Mark R-CNN. Which one is correct? 

Author Response

Comments and Suggestions for Authors

This manuscript explored the automated cardiac chamber size and cardiac physiology of water fleas which are important invertebrates. The authors performed well in the experimental design. The authors also mentioned the previous reports and the development of the cardiac physiology measurement of Claocera. This could help more understanding of the history and development of cardiac physiology measurement of Claocera.

For my point of view, it will be better if you can add more details of the impact of this work, such as more application using this automate. 

Thank you for the valuable comment. The author also agree that the addition of the detail impact of the study will benefit the current study. Thus some addition has been done in introduction and conclusion section in the revised manuscript.

Moreover, the authors have to check the correction of Mask RCNN and Mark R-CNN. Which one is correct? 

Thank you for the comment. The authors already check the corrected term for this and Mask RCNN is the correct one which refer to a deep neural network that used in this study to perform instance segmentation in machine learning. However, the authors also observed that there was two orthographies to describe this term in the manuscript which is Mask RCNN and Mask R-CNN that can potentially confusing for the reader. Although both of it refer to the same thing, the author decide to use Mask RCNN and revise the whole manuscript to make it consistent.

Reviewer 2 Report

I have read the manuscript by Saputra et al. which uses a novel convolutional network approach to estimate cardiac chamber size and performance in response to an environmental toxin in three species of water fleas. The authors describe their development of an automated, machine-learning approach to the calculation of cardiac chamber volume and heart rate in these species and apply this approach to an ecotoxicology dataset involving exposure to an environmental toxin – the herbicide Roundup.

Overall, I feel that the authors have conducted a robust study with important results for the field of crustacean biology. The authors have done a good job at explaining how their present study validates the use of Mask R-CNN convolutional networks as a tool for measuring cardiac responses in Daphnia spp. The authors also do an excellent job of contextualizing their results and of explaining the benefits of this tool for the larger scientific community. I have only a few, minor, suggestions which I believe necessitate additional consideration before publication.

General Comments:

In general, I believe the manuscript would benefit from an expansion of the Introduction section, particularly with respect to historical methodology for measuring cardiac variables in Daphnia spp. For example, it might be useful to include a few sentences in the Introduction describing how cardiac volume/performance have traditionally been measured in Cladocera as a comparison point for why the approach used in this study is novel and/or superior. This will help to better set the stage for the ‘ground-truthing’ comparisons referred to in the Results/Figures.

Specific Comments:

In the Abstract the program “Mask R-CNN” / “Mask RCNN” is written with two different orthographies and later it also appears as “Mask-RCNN”. Suggest using one form consistently throughout the text.

Line 43: Suggest changing “belong” to “belongs” or else adding a category after the word “Most” (e.g., “Most members of the order …”). Also suggest changing “which principally comprise the” to “which is principally comprised of the”.

Line 94: Change “foods” to “food”.

Author Response

Comments and Suggestions for Authors

I have read the manuscript by Saputra et al. which uses a novel convolutional network approach to estimate cardiac chamber size and performance in response to an environmental toxin in three species of water fleas. The authors describe their development of an automated, machine-learning approach to the calculation of cardiac chamber volume and heart rate in these species and apply this approach to an ecotoxicology dataset involving exposure to an environmental toxin – the herbicide Roundup.

Overall, I feel that the authors have conducted a robust study with important results for the field of crustacean biology. The authors have done a good job at explaining how their present study validates the use of Mask R-CNN convolutional networks as a tool for measuring cardiac responses in Daphnia spp. The authors also do an excellent job of contextualizing their results and of explaining the benefits of this tool for the larger scientific community. I have only a few, minor, suggestions which I believe necessitate additional consideration before publication.

General Comments:

In general, I believe the manuscript would benefit from an expansion of the Introduction section, particularly with respect to historical methodology for measuring cardiac variables in Daphnia spp. For example, it might be useful to include a few sentences in the Introduction describing how cardiac volume/performance have traditionally been measured in Cladocera as a comparison point for why the approach used in this study is novel and/or superior. This will help to better set the stage for the ‘ground-truthing’ comparisons referred to in the Results/Figures.

Thank you for the wonderful suggestion. Although the previously used method to calculate the cardiac performance parameter was already described in discussion section, the author strongly agree with the reviewer suggestion as the addition of previous calculation method will be more fit in the introduction section as it will set a better stage for the importance of the study. Thus the revised manuscript was already updated according to the reviewer suggestion.

Specific Comments:

In the Abstract the program “Mask R-CNN” / “Mask RCNN” is written with two different orthographies and later it also appears as “Mask-RCNN”. Suggest using one form consistently throughout the text.

Thank you for the comment. The authors already check and both of them are refer to the same thing. However the authors strongly agree that only one term should be used to make it consistent, as it could potentially confusing for the reader. Thus the author decide to use Mask RCNN as the term used in this study and revise the whole manuscript according to the comment to make it consistent.

Line 43: Suggest changing “belong” to “belongs” or else adding a category after the word “Most” (e.g., “Most members of the order …”). Also suggest changing “which principally comprise the” to “which is principally comprised of the”.

Thank you for the suggestion. The manuscript has been revised according to the reviewer suggestion.

Line 94: Change “foods” to “food”.

Thank you for the correction. The manuscript has been updated according to the suggestion.

Reviewer 3 Report

The Abstract should do a better job at summarizing the main point of the paper. Mostly, it provides general information, while at the same time not enough detail of the current study. Taking this into consideration, it is useful to rewrite your abstract with your values and main conclusion.

In the abstract, the word sp. should be written vertically, not in italics. Similarly, it should be corrected on lines 248, 262, 274 and Table 1, title of Figures 4, A3; anyway check the whole text.

In introduction, the story could be presented better and more clearly. The messages should come out stronger– that fits with what you have done. Main aims should be written here.

Line 99- The number of H2O must be a subscript.

It will be appropriate to give reference in the calculations to be applied under the Statistical Calculation.

Species names given for the first time should be written long (e.g., Daphnia magna), then shortened (e.g., D. magna). This has been tried to be done in the article, but it should be applied in the whole text.

Author Response

Comments and Suggestions for Authors

The Abstract should do a better job at summarizing the main point of the paper. Mostly, it provides general information, while at the same time not enough detail of the current study. Taking this into consideration, it is useful to rewrite your abstract with your values and main conclusion.

Thank you for the valuable suggestion. The authors also agree that the abstract was lack of detail and did not clearly show the main conclusion of the current study. Thus in the updated manuscript, the abstract has been revised to clearly emphasize the result of the current study.

In the abstract, the word sp. should be written vertically, not in italics. Similarly, it should be corrected on lines 248, 262, 274 and Table 1, title of Figures 4, A3; anyway check the whole text.

Thank you for the thorough review. The author also agree that the word sp. should be written without italic form. Thus the whole manuscript has been revised according to the reviewer suggestion.

In introduction, the story could be presented better and more clearly. The messages should come out stronger– that fits with what you have done. Main aims should be written here.

Thank you for the valuable suggestion. The author also agree that the main focused of this study should be emphasize more in the introduction section. Thus the main focus of the study has been emphasized more in the introduction section on the revised manuscript.

Line 99- The number of H2O must be a subscript.

Thank you for the comment. The author already double check and it seem that the number in H2O was already in subscript form which might missed because of the font used in the manuscript.

It will be appropriate to give reference in the calculations to be applied under the Statistical Calculation.

Thank you for the wonderful suggestion. The author also agree that the statistical calculation section need to be revised and appropriate reference was needed to be added to give a better understanding about the chosen statistical test. Thus the statistical calculation section has been updated according to the reviewer suggestion.

Species names given for the first time should be written long (e.g., Daphnia magna), then shortened (e.g., D. magna). This has been tried to be done in the article, but it should be applied in the whole text.

Thank you for the thorough review. The author also strongly agree with the reviewer comment that the shortened species name should be applied to the whole document after the first time it use. Thus the manuscript has been updated according to the reviewer suggestion.